# ADAPTIVE ADVERSARIAL IMITATION LEARNING

## ABSTRACT

We present the ADaptive Adversarial Imitation Learning (ADAIL) algorithm for learning adaptive policies that can be transferred between environments of varying dynamics, by imitating a small number of demonstrations collected from a single source domain. This is an important problem in robotic learning because in real world scenarios 1) reward functions are hard to obtain, 2) learned policies from one domain are difficult to deploy in another due to varying source to target domain statistics, 3) collecting expert demonstrations in multiple environments where the dynamics are known and controlled is often infeasible. We address these constraints by building upon recent advances in adversarial imitation learning; we condition our policy on a learned dynamics embedding and we employ a domain-adversarial loss to learn a dynamics-invariant discriminator. The effectiveness of our method is demonstrated on simulated control tasks with varying environment dynamics and the learned adaptive agent outperforms several recent baselines.

## 1 INTRODUCTION

Humans and animals can learn complex behaviors via imitation. Inspired by these learning mechanisms, Imitation Learning (IL) has long been a popular method for training autonomous agents from human-provided demonstrations. However, human and animal imitation differs markedly from commonly used approaches in machine learning. Firstly, humans and animals tend to imitate the *goal* of the task rather than the particular motions of the demonstrator (Baker et al., 2007). Secondly, humans and animals can easily handle imitation scenarios where there is a shift in embodiment and dynamics between themselves and a demonstrator. The first feature of human IL can be represented within the framework of Inverse Reinforcement Learning (IRL) (Ng et al., 2000; Abbeel & Ng, 2004; Ziebart et al., 2008), which at a high level casts the problem of imitation as one of matching *outcomes* rather than *actions*. Recent work in adversarial imitation learning (Ho & Ermon, 2016; Finn et al., 2016) has accomplished this by using a discriminator to judge whether a given behavior is from an expert or imitator, and then a policy is trained using the discriminator expert likelihood as a reward. While successful in multiple problem domains, this approach makes it difficult to accommodate the second feature of human learning: imitation across shifts in embodiment and dynamics. This is because in the presence of such shifts, the discriminator may either simply use the embodiment or dynamics to infer whether it is evaluating expert behavior, and as a consequence fails to provide a meaningful reward signal.

In this paper we are concerned with the problem of learning adaptive policies that can be transferred to environments with varying dynamics, by imitating a small number of expert demonstrations collected from a single source domain. This problem is important in robotic learning because it is better aligned with real world constraints: 1) reward functions are hard to obtain, 2) learned policies from one domain are hard to deploy to different domains due to varying source to target domain statistics, and 3) the target domain dynamics oftentimes changes while executing the learned policy. As such, this work assumes ground truth rewards are not available, and furthermore we assume that expert demonstrations come from only a single domain (i.e. an instance of an environment where dynamics cannot be exactly replicated by the policy at training time). To the best of our knowledge, this is the first work to tackle this challenging problem formulation.

Our proposed method solves the above problem by building upon the GAIL (Ho & Ermon, 2016; Finn et al., 2016) framework, by firstly conditioning the policy on a learned dynamics embedding ("context variable" in policy search literature (Deisenroth et al., 2013)). We propose two embedding

approaches on which the policy is conditioned, namely, a direct supervised learning approach and a variational autoencoder (VAE) (Kingma & Welling, 2013) based unsupervised approach. Secondly, to prevent the discriminator from inferring whether it is evaluating the expert behavior or imitator behavior purely through the dynamics, we propose using a Gradient Reversal Layer (GRL) to learn a dynamics-invariant discriminator. We demonstrate the effectiveness of the proposed algorithm on benchmark Mujoco simulated control tasks.

The main contributions of our work include: 1) present a general and novel problem formulation that is well aligned with real world scenarios in comparison to recent literature 2) devise a conceptually simple architecture that is capable of learning an adaptive policy from a small number of expert demonstrations (order of 10s) collected from only one source environment, 3) design an adversarial loss for addressing the covariate shift issue in discriminator learning.

## 2   RELATED WORK

Historically, two main avenues have been heavily studied for imitation learning:  1) Behavioral Cloning (BC) and 2) Inverse Reinforcement Learning (IRL). Though conceptually simple, BC suffers from compound errors caused by covariate shift, and subsequently, often requires a large quantity of demonstrations (Pomerleau, 1989), or access to the expert policy (Ross et al., 2011) in order to recover a stable policy.

Recent advancements in imitation learning (Ho & Ermon, 2016; Finn et al., 2016) have adopted an adversarial formation that interleaves between 1) discriminating the generated policy against the expert demonstrations and 2) a policy improvement step where the policy aims to fool the learned discriminator.

Dynamics randomization (Tobin et al., 2017; Sadeghi & Levine, 2016; Mandlekar et al., 2017; Tan et al., 2018; Pinto et al., 2017; Peng et al., 2018; Chebotar et al., 2018; Rajeswaran et al., 2016) has been one of the prevailing vehicles for addressing varying simulation to real-world domain statistics. This avenue of methods typically involves perturbing the environment dynamics (often times adversarially) in simulation in order to learn an adaptive policy that is robust enough to bridge the "Reality Gap".

While dynamics randomization has been explored for learning robust policies in an RL setting, it has a critical limitation in the imitation learning context: large domain shifts might result in directional differences in dynamics, therefore, the demonstrated actions might no longer be admissible for solving the task in the target domain. Our method (Figure 2) also involves training in a variety of environments with different dynamics. However, we propose conditioning the policy on an explicitly learned dynamics embedding to enable adaptive policies based on online system ID.

Yu et al. (2017) adopted a similar approach towards building adaptive policies. They learn an online system identification model and condition the policy on the predicted model parameters in an RL setting. In comparison to their work, we do not assume access to the ground truth reward signals or the ground truth physics parameters at evaluation time, which makes this work's problem formulation a harder learning problem, but with greater potential for real-world applications. We will compare our method with Yu et al. (2017) in the experimental section.

Third person imitation learning (Stadie et al., 2017) also employs a GRL (Ganin & Lempitsky, 2014) under a GAIL-like formulation with the goal of learning expert behaviors in a new domain. In comparison, our method also enables learning adaptive policies by employing an online dynamics identification component, so that the policies can be transferred to a class of domains, as opposed to one domain. In addition, learned policies using our proposed method can handle online dynamics perturbations.

Meta learning (Finn et al., 2017) has also been applied to address varying source to target domain dynamics (Duan et al., 2017; Nagabandi et al., 2018). The idea behind meta learning in the context of robotic learning is to learn a meta policy that is "initialized" for a variety of tasks in simulation, and then fine-tune the policy in the real-world setting given a specific goal. After the meta-learning phase, the agent requires significantly fewer environment interactions to obtain a policy that solves the task. In comparison to meta learning based approaches, fine-tuning on the test environment is

not required in our method, with the caveat being that this is true only within the target domain where the dynamics posterior is effective.

## 2.1 BACKGROUND

In this section we will briefly review GAIL (Ho & Ermon, 2016). Inspired by GANs, the GAIL objective is defined as:

$$\min_\theta \max_\omega \mathbf{E}_{\pi_E}[\log D_\omega(s,a)] + \mathbf{E}_{\pi_\theta}[\log(1 - D_\omega(s,a))] \tag{1}$$

Where $\pi_E$ denotes the expert policy that generated the demonstrations; $\pi_\theta$ is the policy to imitate the expert; $D$ is a discriminator that learns to distinguish between $\pi_\theta$ and $\pi_E$ with generated state-action pairs. In comparison to GAN optimization, the GAIL objective is rarely differentiable since differentiation through the environment step is often intractable. Optimization is instead achieved via RL-based policy gradient algorithms, e.g., PPO (Schulman et al., 2017) or off policy methods, e.g., TD3 (Kostrikov et al., 2018). Without an explicit reward function, GAIL relies on reward signals provided by the learned discriminator, where a common reward formulation is $r_\omega(s,a) = -\log(1 - D_\omega(s,a))$.

## 3 ADAPTIVE ADVERSARIAL IMITATION LEARNING (ADAIL)

### 3.1 PROBLEM DEFINITION

Suppose we are given a class $E$ of environments with different dynamics but similar goals, a domain generator $g(c)$ which takes in a code $c$ and generates an environment $e_c \in E$, and a set of expert demonstrations $\{\tau_{exp}\}$ collected from one source environment $e_{exp} \in E$. In adaptive imitation learning, one attempts to learn an adaptive policy $\pi_\theta$ that can generalize across environments within $E$. We assume that the ground truth dynamics parameters $c$, which are used to generate the simulated environments, are given (or manually sampled) during the training phase.

### 3.2 ALGORITHM OVERVIEW

We allow the agent to interact with a class of similar simulated environments with varying dynamics parameters, which we call "adaptive training". To be able to capture high-level *goals* from a small set of demonstrations, we adopt a approach similar to GAIL. To provide consistent feedback signals during training across environments with different dynamics, the discriminator should be dynamics-invariant. We enable this desirable feature by learning a dynamics-invariant feature layer for the discriminator by 1) adding another head $D^R(c|s,a)$ to the discriminator to predict the dynamics parameters, and 2) inserting a GRL in-between $D^R$ and the dynamics-invariant feature layer. The discriminator design is illustrated in Figure 1. In addition, to enable adaptive policies, we introduced a dynamics posterior that takes a roll-out trajectory and outputs an embedding, on which the policy is conditioned. Intuitively, explicit dynamics latent variable learning endows the agent with the ability to identify the system and act differently against changes in dynamics. Note that a policy can learn to infer dynamics implicitly, without the need for an external dynamics embedding. However, we find experimentally that policies conditioned explicitly on the environment parameters outperform those that do not. The overall architecture is illustrated in Figure 2. We call the algorithm Adaptive Adversarial Imitation Learning (ADAIL), with the following objective (note that for brevity, we for now omit the GRL term discussed in Section 3.4):

$$\min_\theta \max_{\omega,\phi} \mathbf{E}_{\pi_E}[\log D_\omega(s,a)] + \mathbf{E}_{\pi_\theta(\cdot|c)}[\log(1 - D_\omega(s,a))] + \mathbf{E}_{\tau \sim \pi_\theta(\cdot|c)}[\log Q_\phi(c|\tau)] \tag{2}$$

Where $c$ is a learned latent dynamics representation that is associated with the rollout environment in each gradient step; $\tau$ is a roll-out trajectory using $\pi_\theta(\cdot|c)$ in the corresponding environment; $Q(c|\tau)$ is a "dynamics posterior" for inferring the dynamics during test time; The last term in the objective, $\mathbf{E}_{\tau \sim \pi_\theta(\cdot|c)}[\log Q_\phi(c|\tau)]$, is a general form of the expected log likelihood of $c$ given $\tau$. Note that, the posterior training is on-policy, meaning that the rollouts are collected online using the current

---

**Algorithm 1** ADAIL

---

1: **Inputs:**
2: An environment class $E$.
3: Initial parameters of policy $\theta$, discriminator $\omega$, and posterior $\phi$.
4: A set of expert demonstrations $\{\tau_{exp}\}$ on one of the environment $e_{exp} \in E$. An environment generator $g(c)$ that takes a code and generates an environment $e_c \in E$. A prior distribution of $p(c)$.
5: **for** i = 1, 2, .. **do**
6:     Sample $c \sim p(c)$ and generate environment $e_c \sim g(c)$
7:     Sample trajectories $\tau_i \sim \pi_\theta(\cdot|c)$ in $e_c$ and $\tau_i^e \sim \{\tau_{exp}\}$
8:     Update the discriminator parameters $\omega$ with the gradients: $\hat{E}_{(s,a)\sim\tau_i}[\nabla_w \log(D_w(s,a))] + \hat{E}_{(s,a)\sim\tau_i^e}[\nabla_w \log(1 - D_w(s,a))]$
9:     Update the discriminator parameters $\omega$ again with the following loss, such that the gradients are reversed when back-prop through the dynamics-invariant layer: $-\hat{E}_{(s,a)\sim\tau_i}[\log(D^R(c|s,a))]$
10:     Update the posterior parameters $\phi$ with gradients $\hat{E}_{\tau_i}[\nabla_\phi \log Q_\phi(c|\tau_i)]$
11:     Update policy $\pi_\theta(\cdot|c)$ using policy optimization method (PPO) with: $\hat{E}_{(s,a)\sim\tau_i}[-\log(1 - D_\omega(s,a))]$
12: **Output:** Learned policy $\pi_\theta$, and posterior $Q_\phi$.

---

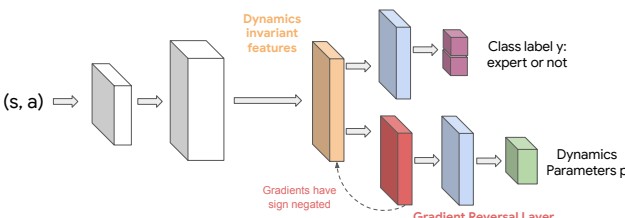

Figure 1: Discriminator with Gradients Reversal Layer (GRL). The red layer is the GRL which reverses the gradients during backprop. The yellow layer is a dynamics-invariant layer that is shared with the classification task.

policy, thereby the last term of the objective is dependent on $\theta$. One can employ various supervised and unsupervised methods towards optimizing this term. We will explore a few methods in the following subsections.

The algorithm is outlined in Algorithm 1.

## 3.3 ADAPTIVE TRAINING

*Adaptive training* is achieved through 1) allowing the agent to interact with a class of similar simulated environments within class $E$, and 2) learning a dynamics posterior for predicting the dynamics based on rollouts. The environment class $E$ is defined as a set of parameterized environments with $n$ degrees of freedom, where $n$ is the total number of latent dynamics parameters that we can change. We assume that we have access to an environment generator $g(c)$ that takes in a sample of the

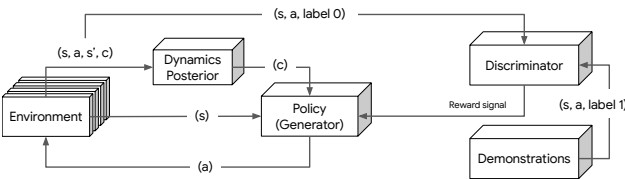

Figure 2: The ADAIL architecture. "Environment" is sampled from a population of environments with varying dynamics, "Demonstrations" are collected from *one* environment within the environment distribution, "Posterior" is the dynamics predictor, $Q(c|\tau)$; Latent code "$c$" represents the ground truth or learned dynamics parameters; The policy input is extended to include the latent dynamics embedding $c$.

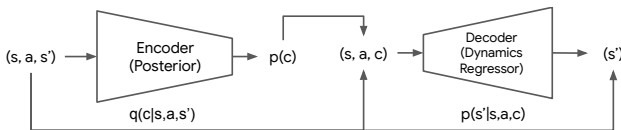

Figure 3: VAE-based unsupervised dynamics latent variable learning.

dynamics parameters $c$ and generates an environment. At each time when an on-policy rollout is initiated, we re-sample the dynamics parameters $c$ based on a predefined prior distribution $p(c)$.

## 3.4 LEARNING A DYNAMICS-INVARIANT DISCRIMINATOR

GAIL learns from the expert demonstrations by matching an implicit state-action occupancy measure. However, this formulation might be problematic in our training setting, where on-policy rollouts are collected from environments with varying dynamics. In non-source environments, the discriminator can no longer provide canonical feedback signals. This motivates us to learn a *dynamics-invariant* feature space, where, the behavior-oriented features are preserved but dynamics-identifiable features are removed. We approach this problem by assuming that the behavior-oriented characteristics and dynamics-identifiable characteristics are loosely coupled and thereby we can learn a dynamics-invariant representation for the discriminator. In particular, we employ a technique called a Gradient Reversal Layer (GRL) (Ganin & Lempitsky, 2014), which is widely used in image domain adaptation (Bousmalis et al., 2016). The dynamics-invariant features layer is shared with the original discriminator classification head, illustrated in Figure 1.

## 3.5 DIRECT SUPERVISED DYNAMICS LATENT VARIABLE LEARNING

Perhaps one of the best latent representations of the dynamics is the ground truth physics parameterization (gravity, friction, limb length, etc). In this section we explore supervised learning for inferring dynamics. A neural network is employed to represent the dynamics posterior, which is learned via supervised learning by regressing to the ground truth physics parameters given a replay buffer of policy rollouts. We update the regression network using a Huber loss to match environment dynamics labels. Details about the Huber loss can be found in appendix A.1. During training, we condition the learned policy on the *ground truth* physics parameters. During evaluation, on the other hand, the policy is conditioned on the *predicted* physics parameters from the posterior.

We use (state, action, next state) as the posterior's input, i.e., $Q_\phi(c|s, a, s')$, and a 3-layer fully-connected neural network to output the N-dimensional environment parameters. Note that one can use a recurrent neural network and longer rollout history for modeling complex dynamic structures, however we found that this was not necessary for the chosen evaluation environments.

## 3.6 VAE-BASED UNSUPERVISED DYNAMICS LATENT VARIABLE LEARNING

For many cases, the number of varying latent parameters of the environment is high, one might not know the set of latent parameters that will vary in a real world laboratory setting, or the latent parameters are oftentimes strongly correlated (e.g., gravity and friction) in terms of their effect on environment dynamics. In this case, predicting the exact latent parameterization is hard. The policy is mainly concerned with the end effector of the latent parameters. This motivates us to use a unsupervised tool to extract a latent dynamics embedding. In this section, we explore a VAE-based unsupervised approach similar to conditional VAE (Sohn et al., 2015) with an additional contrastive regularization loss, for learning the dynamics without ground truth labels.

With the goal of capturing the underlying dynamics, we avoid directly reconstructing the (state, action, next state) tuple, $(s, a, s')$. Otherwise, the VAE would likely capture the latent structure of the state space. Instead, the decoder is modified to take-in the state-action pair, $(s, a)$, and a latent code, $c$, and outputs the next state, $s'$. The decoder now becomes a forward dynamics predictive model. The unsupervised dynamics latent variable learning method is illustrated in Figure 3.

The evidence lower bound (ELBO) used is:

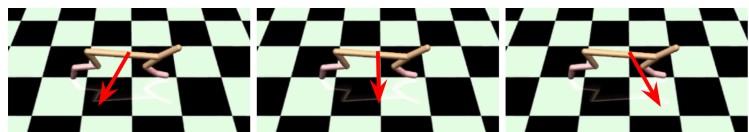

Figure 4: Vary x-component of gravity in HalfCheetah environment. The red arrows in the picture show the gravity directions.

$$\mathbf{ELBO} = \mathbf{E}_{Q_\phi(c|s,a,s')}[\log P_\psi(s'|s,a,c)] - \mathbf{KL}(Q_\phi(c|s,a,s')||P(c)) \tag{3}$$

Where $Q(c|s,a,s')$ is the dynamics posterior (encoder); $P(s'|s,a,c)$ is a forward dynamics predictive model (decoder); $P(c)$ is a Gaussian prior over the latent code $c$. Similar to Davis et al. (2007) and Hsu & Kira (2015), to avoid the encoder learning an identity mapping on $s'$, we add the following KL-based contrastive regularization to the loss:

$$\mathbf{L}_{contrastive} = \mathbf{KL}(Q_\phi(s_0,a_0,s_0')||Q_\phi(s_1,a_1,s_1')) - \min\{\mathbf{KL}(Q_\phi(s_2,a_2,s_2')||Q_\phi(s_3,a_3,s_3')), D_0\}$$

Where $(s_0,a_0,s_0')$ and $(s_1,a_1,s_1')$ are sampled from the same roll-out trajectory; $(s_2,a_2,s_2')$ and $(s_3,a_3,s_3')$ are sampled from different roll-out trajectories. $D_0$ is a constant. We use this regularization to introduce additional supervision in order to improve the robustness of the latent posterior.

The overall objective for the dynamics learner is

$$\min_{\phi,\psi} -\mathbf{ELBO} + \lambda\mathbf{L}_{contrastive} \tag{4}$$

where $\lambda$ is a scalar to control the relative strength of the regularization term. The learned posterior (encoder) infers the latent dynamics, which is used for conditioning the policy. The modified algorithm can be found in the appendix (Algorithm 2).

## 4 EXPERIMENTS

### 4.1 ENVIRONMENTS

To evaluate the proposed algorithm we consider 4 simulated environments: CartPole, Hopper, HalfCheetah and Ant. The chosen dynamics parameters are specified in table 3, and an example of one such parameter (HalfCheetah gravity component $x$) is shown in Figure 4. During training the parameters are sampled uniformly from the chosen range. Source domain parameters are also given in the table 3. For each source domain, we collect **16** expert demonstrations.

**Gym CartPole-V0**: We vary the force magnitude in continuous range $[-1, 1]$ in our training setting. Note that the force magnitude can take negative values, which flips the force direction.

**3 Mujoco Environments: Hopper, HalfCheetah, and Ant**: With these three environments, we vary 2d dynamics parameters: gravity x-component and friction.

| Environment | Paramater 1 | | Parameter 2 | | Source |
|---|---|---|---|---|---|
| CartPole-V0 | $Fm$ | [-1,1] | | | $Fm = 1.0$ |
| Hopper | $Gx$ | [-1.0, 1.0] | $Fr$ | [1.5, 2.5] | $Gx = 0.0, Fr = 2.0$ |
| HalfCheetah | $Gx$ | [-3.0, 3.0] | $Fr$ | [0.0, 2.0] | $Gx = 0.0, Fr = 0.5$ |
| Ant | $Gx$ | [-5.0, 5.0] | $Fr$ | [0.0, 4.0] | $Gx = 0.0, Fr = 1.0$ |

Table 1: Environments. $Fm$ = Force magnitude; $Gx$=Gravity x-component; $Fr$ = Friction. For each environment, we collect **16** expert demonstrations from the source domain.

## 4.2 ADAIL ON SIMULATED CONTROL TASKS

**Is the dynamics posterior component effective under large dynamics shifts?**

We first demonstrate the effectiveness of the dynamics posterior under large dynamics shifts on a toy Gym environment, Cartpole, by varying 1d force magnitude. As the direction of the force changes, blindly mimicking the demonstrations collected from the source domain ($Fm = 1.0$) would not work on target domains with $Fm < 0.0$. This result is evident when comparing ADAIL to GAIL with dynamics randomization. As shown in Figure 5a, GAIL with Dynamics Randomization failed to generalize to $Fm < 0.0$, whereas, ADAIL is able to achieve the same performance as $Fm > 0.0$. We also put a comparison with ADAIL-rand, where the policy is conditioned on uniformly random values of the dynamics parameters, which completely breaks the performance across the domains.

**How does the GRL help improve the robustness of performance across domains?**

To demonstrate the effectiveness of GRL in the adversarial imitation learning formulation, we do a comparative study with and without GRL on GAIL with dynamics randomization in the Hopper environment. The results are shown in Figure 5b.

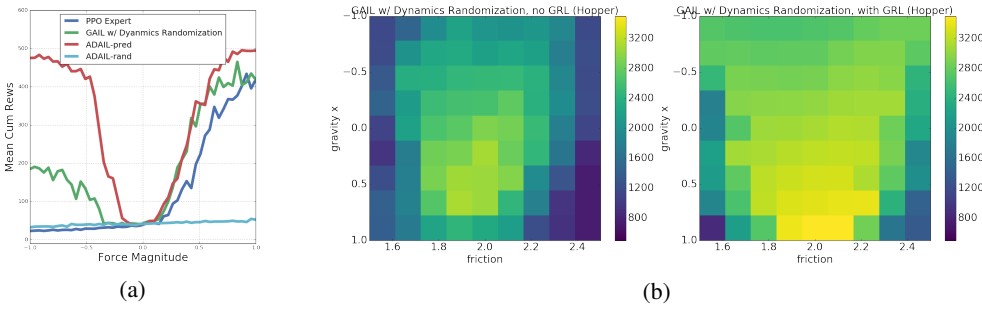

(a)                                                                                  (b)

Figure 5: **(a):** ADAIL on CartPole-V0. Blue: PPO Expert; green: GAIL with Dynamics Randomization; red: ADAIL with latent parameters from the dynamics posterior; light blue: ADAIL with uniformly random latent parameters. **(b):** GAIL with Dynamics Randomization without (left, $2024.89 \pm 669.39$) or with (right, $2453.63 \pm 430.51$) GRL on Hopper

**How does the overall algorithm work in comparison with baseline methods?**

We demonstrate the overall performance of ADAIL by applying it to three Mujoco control tasks: HalfCheetah, Ant and Hopper. For each of the Mujoco environments, we vary 2 continuous dynamics parameters and we compare the performance of ADAIL with a few baseline methods, including 1) the PPO expert which was used to collect demonstrations; 2) the UP-true algorithm of Yu et al. (2017), which is essentially a PPO policy conditioned on ground truth physics parameters; and 3) GAIL with dynamics randomization, which is unmodified GAIL training on a variety of environments with varying dynamics. The results of this experiment are show in in Figure 6.

**HalfCheetah** The experiments show that 1) as expected the PPO expert (Plot 6a) has limited adaptability to unseen dynamics. 2) UP-true (Plot 6b) achieves similar performance across test environments. Note that since UP-true has access to the ground truth reward signals and the policy is conditioned on ground truth dynamics parameters, the Plot 6b shows an approximate expected upper bound for our proposed method since we do not assume access to reward signals during policy training, or to ground truth physics parameters at policy evaluation time. 3) GAIL with dynamics randomization (Plot **??**) can generalize to some extent, but failed to achieve the demonstrated performance in the source environment (gravity x = 0.0, friction = 0.5) 4) Plots 9f 9g show evaluation of the proposed method ADAIL with policy conditioned on ground truth physics parameters and predicted physics parameters respectively; ADAIL matches the expert performance in the source environment (gravity x = 0.0, friction = 0.5) and generalizes to unseen dynamics. In particular, when the environment dynamics favors the task, the adaptive agent was able to obtain even higher performance (around friction = 1.2, gravity = 2).

**Ant and Hopper**. We again show favorable performance on both Ant and Hopper in Figure 6.

**How does the algorithm generalize to unseen environments?**

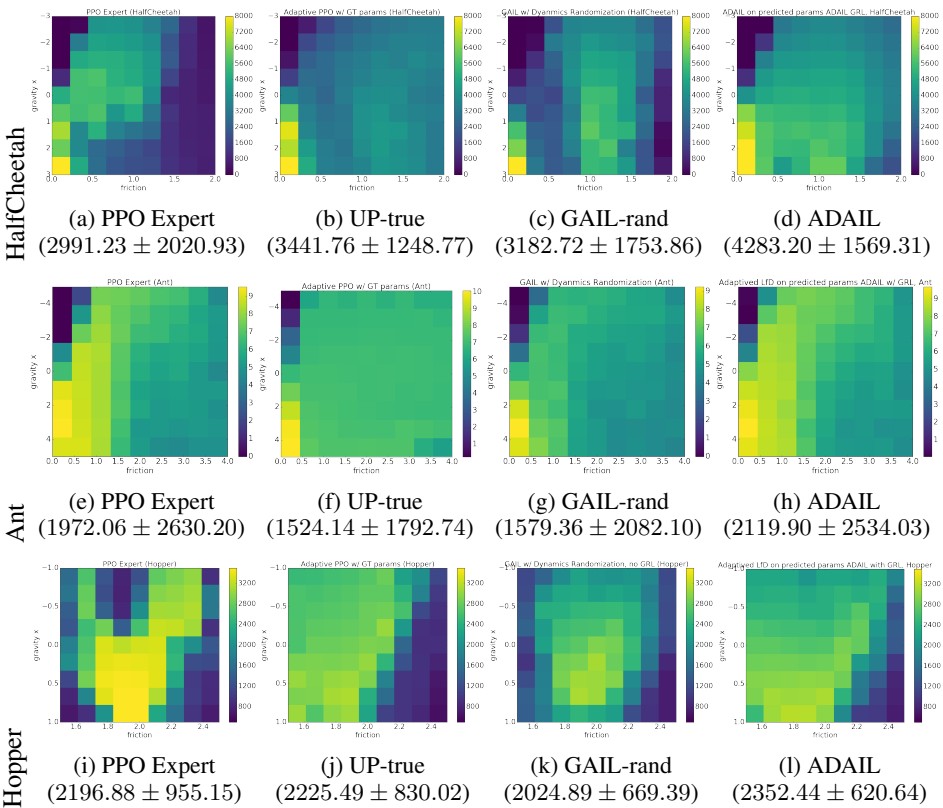

Figure 6: Comparing ADAIL with baselines on Mujoco tasks. Each plot is a heatmap that demonstrates the performance of an algorithm in environments with different dynamics. Each cell of the plot shows 10 episodes averaged cumulative rewards on a particular 2D range of dynamics. Note that to aid visualization, we render plots for Ant in log scale.

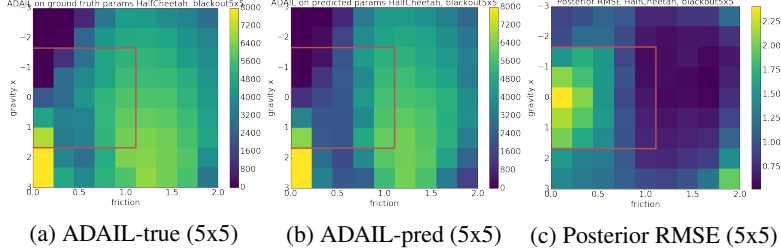

(a) ADAIL-true (5x5)      (b) ADAIL-pred (5x5)      (c) Posterior RMSE (5x5)

Figure 7: Generalization of our policy to held out parameters on the HalfCheetah environment. The red rectangles in plots show the blackout regions not seen during policy training.

To understand how ADAIL generalizes to environments not sampled at training time, we do a suite of studies in which the agent is only allowed to interact in a limited set of environments. Figure 7 shows the performance of ADAIL on different settings, where a $5 \times 5$ region of environment parameters including the expert source environment are "blacked-out". This case is particularly challenging since the policy is not allowed access the domain from which the expert demonstrations were collected, and so our dynamics-invariant discriminator is essential. For additional held out experiments see Appendix A.5.

The experiments show that, 1) without training on the source environment, ADAIL with the ground truth parameters tends to have performance drops on the blackout region but largely is able to generalize (Figure 7a); 2) the posterior's RMSE raises on the blackout region (Figure 7c); 3) consequently ADAIL with the predicted dynamics parameters suffers from the posterior error on the blackout region (Figure 7b).

**How does unsupervised version of the algorithm perform?**

**VAE-ADAIL on HalfCheetah**. With the goal of understanding the characteristics of the learned dynamics latent embedding through the unsupervised method and its impact on the overall algorithm, as a proof of concept we apply VAE-ADAIL to HalfCheetah environment varying a 1D continuous dynamics, friction. The performance is shown in Figure 8.

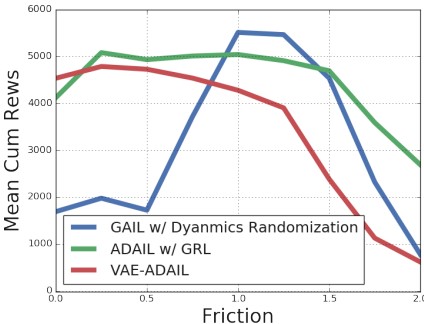

Figure 8: VAE-ADAIL performance on HalfCheetah

## 5    CONCLUSION

In this work we proposed the ADaptive Adversarial Imitation Learning (ADAIL) algorithm for learning adaptive control policies from a limited number of expert demonstrations. We demonstrated the effectiveness of ADAIL on two challenging MuJoCo test suites and compared against recent SoTA. We showed that ADAIL extends the generalization capacities of policies to unseen environments, and we proposed a variant of our algorithm, VAE-ADAIL, that does not require environment dynamics labels at training time. We will release the code to aid in reproduction upon publication.

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

## A  APPENDIX

### A.1  HUBER LOSS FOR DYNAMICS EMBEDDING LOSS

We use the following loss function when training the dynamics embedding posterior:

$$L_\delta(c, Q_\phi(\tau)) = \begin{cases} \frac{1}{2}(c - Q_\phi(\tau))^2 & \text{for } |c - Q_\phi(\tau)| < \delta \\ \delta|c - Q_\phi(\tau)| - \frac{1}{2}\delta^2 & \text{otherwise} \end{cases} \tag{5}$$

Where $\delta$ controls the joint position between L2 and L1 penalty in Huber loss.

**Lemma 1.** Minimizing the above Huber loss is equivalent to maximizing the log likelihood, $\log P(c|\tau)$, assuming $P(c|\tau)$ is distributed as a Gaussian distribution when $|c - Q_\phi(\tau)| < \delta$, and as a Laplace distribution otherwise. See appendix A.2 for the proof.

## A.2 LEMMA 1 PROOF

**Proof.** For $|c - Q_\phi(\tau)| < \delta$,

$$\log P(c|\tau) = \log \frac{1}{\sqrt{2\pi}\sigma_1} e^{-\frac{(c-Q(\tau))^2}{2\sigma_1^2}} \qquad \sigma_1 \text{ is a positive constant} \qquad (6)$$

$$\log P(c|\tau) = \log \frac{1}{\sqrt{2\pi}\sigma_1} - \frac{1}{2\sigma_1^2}(c - Q(\tau))^2 \qquad (7)$$

$$\nabla \log P(c|\tau) = \nabla(\log \frac{1}{\sqrt{2\pi}\sigma_1} - \frac{1}{2\sigma_1^2}(c - Q(\tau))^2) \qquad (8)$$

$$= -C_1 \nabla \frac{1}{2}(c - Q(\tau))^2 \qquad C_1 \text{ is a positive constant} \qquad (9)$$

$$= -C_1 \nabla L_\delta(c, Q_\phi(\tau)) \qquad (10)$$

Likewise, we can prove for $|c - Q_\phi(\tau)| \geq \delta$. ∎

## A.3 VAE-ADAIL ALGORITHM

---

**Algorithm 2** VAE-ADAIL

---

1: **Inputs:**
2: An environment class $E$.
3: Initial parameters of policy $\theta$, discriminator $\omega$, and dynamics posterior $\phi, \psi$.
4: A set of expert demonstrations $\{\tau_{exp}\}$ on one of the environment $e_{exp} \in E$.
5: **for** i = 1, 2, .. **do**
6:     Sample environment $e \in E$.
7:     Sample trajectories $\tau_i \sim \pi_\theta(\cdot|Q_\phi)$ in $e$ and $\tau_i^e \sim \{\tau_{exp}\}$
8:     Update the discriminator parameters $\omega$ with the gradients

$$\hat{E}_{(s,a)\sim\tau_i}[\nabla_w \log(D_w(s, a))] + \hat{E}_{(s,a)\sim\tau_i^e}[\nabla_w \log(1 - D_w(s, a)]$$

9:     Update the posterior parameters $\phi, \psi$ with the objective described in Eq (3) & (4)
10:     Update policy $\pi_\theta(\cdot|c)$ using policy optimization method (TRPO/PPO) with:

$$\hat{E}_{(s,a)\sim\tau_i}[-\log(1 - D_\omega(s, a))]$$

11: **Output:** Learned policy $\pi_\theta$, and posterior $Q_\phi$.

---

## A.4 HALFCHEETAH ADAIL PERFORMANCE COMPARISON

Figure 9: Comparing ADAIL with a few baselines on HalfCheetah. Each plot is a heatmap that demonstrates the performance of an algorithm in environments with different dynamics. Each cell of the plot shows 10 episodes averaged cumulative rewards on a particular 2D range of dynamics.

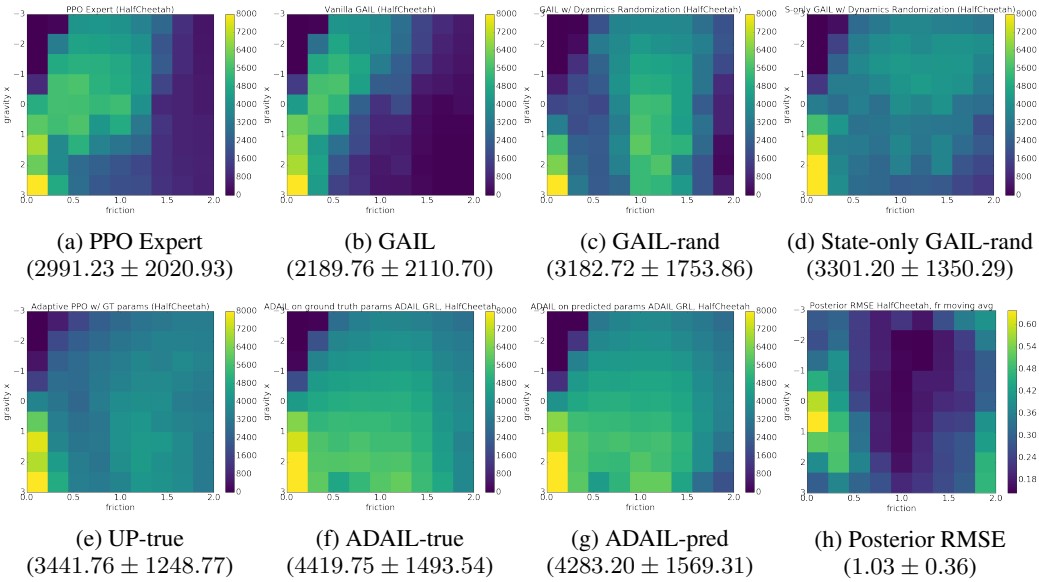

(a) PPO Expert
(2991.23 ± 2020.93)

(b) GAIL
(2189.76 ± 2110.70)

(c) GAIL-rand
(3182.72 ± 1753.86)

(d) State-only GAIL-rand
(3301.20 ± 1350.29)

(e) UP-true
(3441.76 ± 1248.77)

(f) ADAIL-true
(4419.75 ± 1493.54)

(g) ADAIL-pred
(4283.20 ± 1569.31)

(h) Posterior RMSE
(1.03 ± 0.36)

## A.5 HELD-OUT ENVIRONMENT EXPERIMENT

## A.6 HYPERPARAMETERS

### A.6.1 ADAIL

We used fully connected neural networks with 2 hidden layers for all three components of the system. The Network hyperparameters for each of the test environments with 2D dynamics parameters are shown in the following table. For UP-True and GAIL-rand, we use the same set of hyperparameters.

| Environment | Policy | | Discriminator | | Posterior | |
|---|---|---|---|---|---|---|
| | Architecture | Learning rate | Architecture | Learning rate | Architecture | Learning rate |
| CartPole-V0 | (s,a) - 64 - 64 - (a) | 0.0005586 | (s,a) - 32 - 32 - 1 | 0.000167881 | (s,a,s')-76-140-2 | 0.00532 |
| Hopper | (s,a) - 64 - 64 - (a) | 0.000098646 | (s,a) - 32 - 32 - 1 | 0.0000261 | (s,a,s')-241-236-2 | 0.00625 |
| HalfCheetah | (s,a) - 64 - 64 - (a) | 0.00005586 | (s,a) - 32 - 32 - 1 | 0.0000167881 | (s,a,s')-150-150-2 | 0.003 |
| Ant | (s,a) - 64 - 64 - (a) | 0.000047 | (s,a) - 32 - 32 - 1 | 0.000037 | (s,a,s')-72-177-2 | 0.002353 |

Table 2: ADAIL network architectures and learning rates on test environments

### A.6.2 VAE-ADAIL

Here we show the network architectures and learning rates for VAE-ADAIL.

| | Encoder (Posterior) | Decoder | Policy | Discriminator |
|---|---|---|---|---|
| Architecture | (s,a,s') - 200 - 200 - (c) | (s,a,c) - 200 - 200 - (s') | (s,a) - 64 - 64 - (a) | (s,a) - 32 - 32 - (1,c) |
| Learning rate | 0.000094 | 0.000094 | 0.00005596 | 0.000046077 |

Table 3: VAE-ADAIL network architectures and learning rates

