# OpenReview forum: "Adaptive Adversarial Imitation Learning"
_ICLR.cc/2020/Conference — Reject_

### Official Review · AnonReviewer2 · 2019-10-22
**Official Blind Review #2**

**Rating:** 6

**Review:**

The paper describes an approach that combines domain adversarial neural network with generative adversarial imitation learning. In the setup, each environment is dependent on some latent context variable (e.g. physics parameters) through which latent variable dependent policy and latent variable independent discriminators are learned.

I don't think the exact same idea has appeared in existing literature. The authors justifies its connections and differences between third person imitation learning, but it seems that the proposed method bear some similarities to the NeurIPS19 papers below.

https://arxiv.org/pdf/1909.09314.pdf
https://drive.google.com/file/d/1urPE7J5tT8dzoBHSFvZKwNLsQieU706o/view

The following papers also assumed that GAIL-like methods in a meta learning setup, where environments depend on context. Perhaps the difference here is that the discriminator is also trained with a gradient reversal layer, so it encourages the discriminator to not use redundant state information. Also in this paper the source domain only contains demos from one env, which might highlight the importance of the gradient reversal layer.


Questions:

In "dynamics learning", it seems that the inference network learns is mostly the context variable c, wonder if it is better to use terms like "latent variable inference" to avoid confusion.

What does the standard deviation mean in Figure 6? It seems a lot of them are even larger than the mean.

There is little explanation to the VAE-ADAIL experiments -- is it safe to assume most of the experiments require certain knowledge of the latent variable in order to be successful? Maybe some of the arguments about VAE can go to the appendix.

Why would in some cases UP-True performs much worse than ADAIL? In Ant it is even worse than PPO expert.

How would you adapt to unseen environments in ADAIL-pred? I don't see explanations in the text about how this is done. Essentially, how are samples obtained from the environment in order to perform posterior inference?


**Experience Assessment:**

I have published one or two papers in this area.

**Review Assessment: Checking Correctness Of Derivations And Theory:**

I carefully checked the derivations and theory.

**Review Assessment: Checking Correctness Of Experiments:**

I carefully checked the experiments.

**Review Assessment: Thoroughness In Paper Reading:**

I read the paper thoroughly.

---

> ### Author Response · Authors · 2019-11-14
> **Response to Reviewer #2**
>
> We would like to thank the reviewer for the thoughtful review and valuable feedback. We found the summary in your comment accurate and reflective of the purpose and contributions of our work.
>
> We address the questions below:
>
> “In ‘dynamics learning’, it seems that the inference network learns is mostly the context variable c, wonder if it is better to use terms like "latent variable inference" to avoid confusion.”
>
> Thank you for the suggestion, and have updated our text to reflect the new term.
>
> “What does the standard deviation mean in Figure 6? It seems a lot of them are even larger than the mean.”
>
> It means the standard deviation (STD) across all domains evaluated, not the experimental variance of multiple seeds within a single domain. Some STDs are high because of the performance across some domains exhibit large expected variance, e.g., in Ant and HalfCheetah when friction is less than 0.5 (the robot is not expected to obtain high reward, even for an optimal policy, leading to high variance across all domains).
>
> “There is little explanation to the VAE-ADAIL experiments -- is it safe to assume most of the experiments require certain knowledge of the latent variable in order to be successful? Maybe some of the arguments about VAE can go to the appendix.”
>
> The effectiveness of ADAIL is demonstrated in the experiment section given that we are provided with informative dynamics parameters (either ground truth or predicted) at eval time. We presented an experiment of VAE-ADAIL as a proof-of-concept to demonstrate the unsupervised method. We have added references to related works in our updated paper (Section 3.6), and will further clarify the motivation (and/or add new experimental results) in the next paper revision.
>
> “Why would in some cases UP-True performs much worse than ADAIL? In Ant it is even worse than PPO expert.”
>
> It is a meaningful observation. Figure 6f reflects a tendency of optimizing average performance across domains using UP-True, for which the true reward signals are provided. An interpretation of this result could be that PPO is trained only in one source domain so it is more “focused” to obtain good performance in that domain, so it might be able to obtain better policy in that domain but (as shown empirically) fails to generalize to other domains. Another angle could be that in UP-True, the policy network is shared across domains so policy updates are done using rollouts from multiple domains. Therefore it is not able to "overfit" to a specific domain as the PPO expert does on the source domain. Hence the performance difference.
>
> Since ADAIL imitates demonstrations from the PPO expert (as opposed to learning from scratch), it is able to emulate the performance in the source domain. GRL in this case helps because it mitigates the problem of inconsistent reward signals from the discriminator across shifted domains.
>
> “How would you adapt to unseen environments in ADAIL-pred? I don't see explanations in the text about how this is done. Essentially, how are samples obtained from the environment in order to perform posterior inference?”
>
> As shown in Section 4.2, we demonstrate ADAIL performance to unseen environments (not seen during training). The experimental procedure is as follows:
>
> In a new environment, the posterior performs online inference on each time step given the current (s, a, s’) tuple. The predicted code is subsequently fed to the policy as the conditioned dynamics code. We have also tried moving average style filtering on the posterior predictions with minor performance gain. Thus we omitted multi-step posterior inference.
>
> Comparison with related papers:
>
> Thanks for pointing us to the related recent work! We will make sure to update the related work section.

---

### Official Review · AnonReviewer1 · 2019-10-23
**Official Blind Review #1**

**Rating:** 3

**Review:**

Summary:
The submission considers the problem of imitation learning when the dynamics of the expert are not known to the agent and the dynamics of the agent may change frequently. It is however assumed that the agent has access to a parameterized simulator that can simulate the expert dynamics. The parameters for the simulator are not known but are assumed to be drawn from a known distribution.
The proposed method is based on GAIL but uses several modifications:
- A contextual policy is trained that also takes the dynamics-parameters as additional input. At each iteration, a new environment is sampled for performing the policy-rollout.
- A "posterior" prediction network is trained to maximize the likelihood of the parameters that were used for the different roll-outs. This network is used for the test case, where the true dynamics of the agent are not known.
- The discriminator might use features of the state-action input that correlate with the corresponding dynamics. Classifying based on such features may be undesirable because the discriminator might no longer produce useful rewards. In order to address this problem, an additional head is added to the discriminator that outputs a prediction of the dynamic parameters. The prediction error is trained by backpropagation, however by flipping the sign of the gradient at the last shared layer, the features of the discriminator are optimized to be unsuited for predicting the dynamic parameters (the technique is known as Gradient Reversal Layer).
- A VAE-based method for learning latent dynamic parameters is proposed, by training a conditional VAE to reconstruct the next state, where the current state and action are provided as context to the encoder and decoder.

Contribution:
One of the strong-points of the submission is the fact that it features several different, orthogonal contributions. I also think that the considered problem setting is relatively interesting. However, also when considering real applications such as robotics, I am not convinced that explicitly modeling the dynamic changes is necessary. Some existing imitation learning methods focus on a setting where the dynamics of the expert may differ from the agent, but the dynamics of agents do not change. This setting does not require dynamic-contextual policies and seems to be applicable to typical robot applications.

Soundness:
The different components of the proposed methods seem reasonable to me. They do not come with (and arguably do not require) new derivations but seem rather like pragmatic solutions for the encountered problem.
The optimization problem (Eq.2) seems to be formulated slightly inaccurate, because the last term should in my opinion not depend on theta. If I understand correctly, the policy should not maximize the likelihood of the dynamics posterior.
The contrastive regularization loss needs to be better motivated. A high KL in the first term may not necessary be bad, for example, if the confidence in the prediction of (s_0, a_0, s'_0) is lower compared to (s_1, a_1, s'_1). If a similar regularizer has been used in prior work, such work should be referenced. Otherwise, it needs to be motivated.

Presentation/Clarity:
The presentation of the work is arguably the main weakness of the paper.
The submission does not seem polished. It contains a large amount of typos. Figure 2 is confusing and adds little compared to the text description. Also the structure could be improved. For example, the submission introduces the posterior loss and outlines the algorithm before describing the individual components.
The paper uses some techniques such as conditional VAEs [1] or contextual policy search [2]
are used but not described / referenced.

Evaluation:
I like that the different aspects of the proposed method are also evaluated individually. The ablations with respect to the adaptability and GRL are crucial. The evaluation of the performance could be improved. PPO and UP-True use the true reward function, so the only real competitor is a naive GAIL-baseline that uses randomized dynamics during training. I'm not aware of prior work that considers the exact same setting as the manuscript. However, one of the main arguments for inverse reinforcement learning is the claimed generalizability of a reward function as opposed to a policy. I see that learning a new policy after each change in the dynamics may be too costly in some settings. However, comparisons to methods such as AIRL that aim to learn reward functions that are robust to changes in the dynamics would be highly interesting.

[1] Sohn Kihyuk, Honglak Lee, and Xinchen Yan. “Learning Structured Output Representation using Deep Conditional Generative Models.” Advances in Neural Information Processing Systems. 2015.
[2] Marc Peter Deisenroth, Gerhard Neumann, and Jan Peters.  A survey on policy search forrobotics.Foundations and Trends in Robotics, 2(1–2):328–373, 2013.

Comparison: How about IRL, e.g. AIRL?
What about state-only GAIL?

Typos:
Equations are not properly integrated into the sentences (missing punctuations)
"domains, as oppose to one domain."
"Inspired by to GANs"
"and generates a environment"
Figure 5a (legend): "dyanmics"
"that can generalized across"
Algorithmbox: "A environment", "and Generate environment"
"is achieved through 1) allowing"
"the policy is mainly concerned with the end-effector of the latent parameters"


Question:
What are the network architectures?

According to line 10 of the algorithmbox only the current trajectory is used for updating the dynamics posterior. Why?

**Experience Assessment:**

I have published one or two papers in this area.

**Review Assessment: Checking Correctness Of Derivations And Theory:**

I assessed the sensibility of the derivations and theory.

**Review Assessment: Checking Correctness Of Experiments:**

I assessed the sensibility of the experiments.

**Review Assessment: Thoroughness In Paper Reading:**

I read the paper at least twice and used my best judgement in assessing the paper.

---

> ### Author Response · Authors · 2019-11-14
> **Response to Reviewer #1**
>
> We would like to thank R1 for the thorough review and insightful feedback!
>
> Regarding the contribution, we first thank R1 for recognizing our contributions and novelty. We also believe that this is an important problem formulation for imitation learning.
>
> “... not convinced that explicitly modeling the dynamic changes is necessary”
>
> As shown experimentally, the dynamics posterior is particularly useful when there are large domain shifts. Figure 5 (a) illustrates this scenario, where the direction of the force is negated in some of the test domains. ADAIL is able to cover both the case of non-directional domain shift and directional domain shift. As a comparison, GAIL with dynamics randomization fails in this setting.
>
> “Some existing imitation learning methods focus on a setting where the dynamics of the expert may differ from the agent, but the dynamics of agents do not change. ...”
>
> Our problem formulation covers the suggested setting, where the dynamics in the target domain does not change at eval time. ADAIL is also effective under online dynamics perturbations. This makes it applicable to real-world robot learning.
>
> “.. the last term should in my opinion not depend on theta.” “According to line 10 of the algorithm box only the current trajectory is used for updating the dynamics posterior. Why?”
>
> Sorry for the confusing notation. We will update this in the next paper version. By including theta we meant that the posterior training is on-policy, which makes it dependent on theta (since the environment samples it is trained on is dependent on theta). In practice the posterior could be trained off-policy with sampled rollouts, however, this might either require an additional reply buffer or requires additional rollouts after the policy training converges. Both amount to implementation complexity. We found on-policy posterior training effective in our test scenarios, hence the design decision.
>
> “The contrastive regularization loss needs to be better motivated”
>
> Thank you for bringing it to our attention and we will add additional clarifying text to the next paper revision. The motivation for the contrastive loss is to introduce additional supervision in order to improve the robustness of the latent posterior. We find that without it, the posterior value is prone to drifting during the course of a rollout (especially for frame transitions where the dynamics are difficult to infer). The addition of this contrastive term adds the simple prior that all transitions within a single rollout have the same posterior value. This matches the fact that we sample the environment dynamics parameters once during a single rollout, and then the parameters are constant throughout. Note that in a real-world robotics application, an extension of this simple loss term would be to ensure that adjacent posterior estimates are similar (rather than constant over an entire trajectory) as it is expected that real-world dynamics exhibit high temporal coherence.
>
> “If a similar regularizer has been used in prior work, such work should be referenced”
>
> We agree that metric loss is commonly used, and KL-like distance measure is also not unprecedented in metric learning ([1], [2]). We have added references to the papers in the updated draft. However, to the best of our knowledge, we are the first to use KL-based metric loss to learn dynamics embeddings, especially in the context of imitation learning.
>
> [1] Davis, Jason V., et al. "Information-theoretic metric learning." 2007.
> [2] Hsu, and Kira. "Neural network-based clustering using pairwise constraints." 2015
>
> Regarding the presentation/clarity, thank you for pointing out presentation related issues. We have fixed all the found typos.
>
> “Figure 2 is confusing and adds little compared to the text description.”
>
> We believe Figure 2 helps better illustrate the system graphically to readers with limited background in this field. We have revised Figure 2 to make it more clear and illustrative in the updated draft.
>
> “.. the structure could be improved. .. ”
>
> We will make sure to further polish the structure and presentation of the paper in the next version.
>
> “The paper uses some techniques such as conditional VAEs [1] or contextual policy search [2]
> are used but not described / referenced.”
>
> Thank you for the relevant citations. We have added references to both these works in the updated draft.
>
> “What are the network architectures?”
>
> We have updated details of network architecture in Appendix in our updated draft.

---

> > ### Author Response · Authors · 2019-11-14
> > **Response to Reviewer #1 (Cont'd)**
> >
> > “Comparison: How about IRL, e.g. AIRL? What about state-only GAIL?”
> >
> > We have appended an experiment using state-only GAIL on HalfCheetah in the updated version of the paper (as updated in Figure 9). State-only GAIL-rand achieves similar performance as GAIL-rand, whereas it exhibits less variance across domains. However, ADAIL still significantly outperforms State-only GAIL-rand. State-only GAIL is a nice baseline method to have, as we have seen a handful of papers in the imitation learning literature that are based on state-only observations (despite a lack of theoretical guarantees for such a regime). It is also reasonable to use AIRL as a baseline as it claims to recover the true reward function, which might be transferable across domains.
> >
> > We will update the experimentation section with additional baselines (State-only GAIL in additional environments, and AIRL) on all environments after the rebuttal phase as it requires some additional time for us to obtain and consolidate new experimental results beyond the timeline of discussions.

---

> > > ### Comment · AnonReviewer1 · 2019-11-15
> > > **Thanks for the revision**
> > >
> > > I think that the presentation has been improved noticeably. I need to take a closer look on the current submission to make up my mind on whether to argue for acceptance or not. Until then, I'll keep the initial rating.

---

### Official Review · AnonReviewer3 · 2019-10-23
**Official Blind Review #3**

**Rating:** 1

**Review:**

This paper proposes an algorithm for imitation of expert demonstrations, in situations where the imitator is acting under a different environment (different dynamics, for instance) than the one used to collect expert demonstrations. The algorithm builds on GAIL with the following modifications – the discriminator is made dynamics-invariant by adding a domain-adversarial loss, and the policy is made to condition on a dynamics context. A separate dynamics posterior network is trained (either supervised or unsupervised) to predict this context at test-time.


I have the following concerns about the paper:

1.	Lack of novelty:
         a.	Learning a dynamics-invariant discriminator with the gradient-reversal-layer was proposed in Stadie et. al (2017). How is this approach different? In particular, what is the new element in Figure 1. and complete Section 3.4?
         b.	Learning a posterior network to predict context codes, and conditioning the policy on those was explored in papers such DIAYN and InfoGAIL.

2.	The proposed algorithm is reliant on the possibility of being able to sample from a distribution of environments (with varying dynamics), and then collect many trajectories in that environment (Line 6-7 in Algorithm 1). This is a severe requirement for real-word scenarios, and somewhat antithetic to the robotics learning motivation given by the authors in the introduction. Moreover, Figure 7 seems to imply that the method doesn’t generalize well to unseen environments, if enough environments can’t be sampled during training time.


Minor comment:

Figure 8. Friction value should not go from -3 to 3. Also, this single result inspires no confidence in the benefit or general applicability of the VAE-based context prediction.


**Experience Assessment:**

I have published one or two papers in this area.

**Review Assessment: Checking Correctness Of Derivations And Theory:**

I assessed the sensibility of the derivations and theory.

**Review Assessment: Checking Correctness Of Experiments:**

I assessed the sensibility of the experiments.

**Review Assessment: Thoroughness In Paper Reading:**

I read the paper thoroughly.

---

> ### Author Response · Authors · 2019-11-14
> **Response to Reviewer #3**
>
> We would like to thank you for the valuable feedback! Below we address the raised concerns and answer reviewer questions:
>
> 1a) Third person imitation learning (TPIL) Stadie et al. 2017 and our work both employed a gradient reversal layer (GRL). We explained the differences between Stadie et al. 2017 and our work in the related work section: 1) Third person imitation learning (TPIL) paper only considers two domains: a source domain where the demonstrations are collected and a target domain where the policy is evaluated, whereas, we consider learning an adaptive policy that is transferable to a class of domains with different dynamics. Compared to TPIL, this is a substantially more challenging setting, with relaxed assumptions and consequently greater potential for real robotic applications. 2) in addition, we employed a dynamics posterior to actively predict the latent dynamics parameters at each time step during evaluation making the policy agnostic to online dynamics perturbations, which is impossible with TPIL.
>
> We would also like to do a comparative analysis with TPIL in the next version of the paper.
>
> “In particular, what is the new element in Figure 1. and complete Section 3.4?”
>
> We use Figure 1 to illustrate our discriminator architecture. In retrospect, we agree with R3 that the description of GRL might be redundant. We have shortened our description of domain adversarial loss in Section 3.4 in the updated version of the manuscript.
>
> 1b) Conditioning the policy on a contextual variable is indeed a common idea that is shared in the literature. Our novelty, however, lies in considering a new problem formulation (which was not explored before), and unifying a number of technical solutions for solving the problem as highlighted by R1.
>
> Thank you for making the connections with DIAYN (Eysenbach et al. 2018), and InfoGAIL (Li et al. 2017). However, we believe that DIAYN and InfoGAIL are solving different problems. We compare the problem formulations of DIAYN, and InfoGAIL with our work to further illustrate the argument:
>
> Diversity is all you need (DIAYN) Eysenbach et al. 2018 seeks to pretrain diversified skills without a given reward function by training a discriminator to discriminate different skills under the same dynamics. DIAYN uses maximum entropy policy to diversify the learned skills. The paper briefly mentioned learning from a few demonstrations of different skills in experiment section. Our method, however, learns an adaptive policy that is transferable under different dynamics.
>
> InfoGAIL Li et al. 2017 considered a setting where the dynamics does not change, and a mixture of demonstrations are collected from multiple experts that exhibit behavioral variations and/or try to achieve different goals. The InfoGAIL algorithm is not suitable for learning transferable policies under domain shifts. Our method, while sharing some architectural similarities to infoGAIL, on the other hand learns an adaptive policy that tries to achieve the same goal under different dynamics with demonstrations collected from a single source domain.
>
>
> 2 “This is a severe requirement for real-word scenarios, ...”
>
> We respectfully disagree that sampling environments from a distribution is an unrealistic assumption. We believe the ability to configure environments with different dynamics is a necessity for learning transferable policies under a model-free RL setting. Previous methods that try to learn transferable policies (Tobin et al., 2017; Sadeghi & Levine, 2016; Mandlekar et al., 2017; Tanet al., 2018; Pinto et al., 2017; Peng et al., 2018; Chebotar et al., 2018; Rajeswaran et al., 2016) are also reliant on the assumption of sampling environments from a distribution or the ability to perturb the environment dynamics; these authors either instrument the environment or policy (e.g. torque applied, slope of ground for locomotion tasks, etc) in order to control environment parameters during policy training, or they model parameter variation explicitly.
>
> As is standard in imitation learning, we do not assume that the parameterization of the optimal policy is known. Likewise, we assume that we cannot control the environment parameters of the expert demonstration, and that the expert rollouts come from a single source environment. To the best of our knowledge, this is the first work to make this relaxed assumption. We believe that this relaxed assumption makes our proposed method more applicable to real robotic learning scenarios than much of the relevant literature.

---

> > ### Author Response · Authors · 2019-11-14
> > **Response to Reviewer #3 (Cont'd)**
> >
> > “Figure 7 seems to imply that the method doesn’t generalize well to unseen environments, if enough environments can’t be sampled during training time”
> >
> > We would like to clarify that the demonstrations in Figure 7 are collected from within the blackout area, as discussed in Section 4.2. Furthermore, the policy is not executed in this region at training time. This experiment demonstrates that ADAIL can generalize without training on these environments with minimal drop in performance when provided with the ground truth dynamics parameters. Figure 7 also shows that the posterior prediction is crucial when evaluated in unseen environment (comparing Figure 7a and Figure 7b). This is the main reason why we witnessed some degraded performance.
> >
> > “Figure 8. Friction value should not go from -3 to 3. “
> >
> > Thanks for pointing it out! It is a typo. We have updated the values from 0 to 2 in the updated version of the draft.
> >
> > “Also, this single result inspires no confidence in the benefit or general applicability of the VAE-based context prediction.”
> >
> > The effectiveness of ADAIL is demonstrated in the experiment section given that we are provided with informative dynamics parameters (either ground truth or predicted) at eval time. We included an experiment of VAE-ADAIL as a proof-of-concept to demonstrate the unsupervised method. We will make sure to add more experiments with the VAE-ADAIL in the next version of the paper.

---

### Author Response · Authors · 2019-11-14
**Summary of changes to the paper**

We would like to thank the reviewers for their time and valuable feedback. We have made the following changes to the manuscript based on the reviews:

1. Added vanilla GAIL and state-only GAIL experiments on HalfCheetah in the Appendix (Figure 9) [As suggested by R1]
2. Added a reference to conditional VAE (Sohn et al., 2015) in Section 3.6 and a reference to contextual policy search (Deisenroth et al., 2013) in Introduction paragraph 3. [As suggested by R1]
3. Added references to previous work of KL-based contrastive loss (Davis et al. 2007 and Hsu & Kira 2015) in Section 3.6
4. Shortened the description of GRL in Section 3.4 [As suggested by R3]
5. Revised Figure 1 and Figure 2
6. Updated term “dynamics learning” to “dynamics latent variables learning” [As suggested by R2]
7. Updated all typos found [As suggested by R1 and R3]

---

### Decision · Program_Chairs · 2019-12-19

**Decision:**

Reject

**Comment:**

This paper extends adversarial imitation learning to an adaptive setting where environment dynamics change frequently. The authors propose a novel approach with pragmatic design choices to address the challenges that arise in this setting. Several questions and requests for clarification were addressed during the reviewing phase. The paper remains borderline after the rebuttal. Remaining concerns include the size of the algorithmic or conceptual contribution of the paper.